# Shared Genetics in Celiac Disease and Inflammatory Bowel Disease Specify a Greater Role for Intestinal Epithelial Cells

**DOI:** 10.3390/ijms26072982

**Published:** 2025-03-25

**Authors:** Nathan Vinícius Ribeiro, Sajid Anwar, Sebo Withoff, Iris H. Jonkers

**Affiliations:** Department of Genetics, University Medical Center Groningen, Antonius Deusinglaan 1, 9713 AV Groningen, The Netherlands; n.v.ribeiro@umcg.nl (N.V.R.); s.anwar@umcg.nl (S.A.); s.withoff@umcg.nl (S.W.)

**Keywords:** celiac disease, inflammatory bowel disease, genetics, epithelial cells

## Abstract

The contribution of genetics to the development of gut-related autoimmune diseases such as celiac disease (CeD) and inflammatory bowel diseases (IBDs) is well-established, especially in immune cells, but pinpointing the significance of genetic variants to other cell types is more elusive. Increasing evidence indicates that intestinal epithelial cells are active players in modulating the immune response, suggesting that genetic variants affecting these cells could change cell behavior during disease. Moreover, fine-mapping genetic variants and causal genes to relevant cell types can help to identify drug targets and develop personalized targeted therapies. In this context, we reviewed the functions of genes in disease-associated loci shared by CeD and IBD that are expressed in epithelial cells and explored their potential impacts.

## 1. Introduction

With the rapid worldwide rise in the incidence of gut-related autoimmune diseases such as celiac disease (CeD) and inflammatory bowel diseases (IBDs), there is an increasing need for novel diagnostic and treatment approaches [1,2]. Gaining a better understanding of the mechanisms underlying their pathology is key to this effort. While dietary gluten (see Appendix A, glossary) acts as the environmental driver for CeD, the drivers for IBD are still unknown. Nevertheless, extensive research has been undertaken to understand the immunopathology of these diseases (Appendix B) and how it is affected by genetic factors. Genome-wide association studies (GWASs) have now firmly established the significance of genetics in CeD and IBD, uncovering hundreds of single nucleotide polymorphisms (SNPs) associated to these diseases. However, 90% of associated genetic variants are located in the non-coding genome, where they could influence gene expression in specific cell types through disruptions in gene regulatory elements such as promoters or enhancers [3]. To date, the role of these SNPs in disease is not well understood, nor is it clear in which cell types these SNPs are relevant.

Thus far, SNPs associated with autoimmune diseases have most often been studied in immune cells, but they could also affect other cell types, including the epithelial cells that make up the physical barrier along the gastrointestinal tract that is affected in CeD and IBD. This barrier defends against potentially harmful molecules and microorganisms and manages the uptake of essential nutrients and solutes (Appendix C). Barrier disruption may lead to the increased influx of gluten peptides, a CeD hallmark, or increased exposure to harmful microbial components, as in IBD, which may increase the risk of disease onset [4,5]. In both CeD and IBD, immune cell infiltration, microbiome dysbiosis, and an increased influx of proinflammatory external epitopes are hallmarks of disease [6]. However, it remains unclear whether barrier defects are a cause or consequence of autoimmunity. In this review, we outline possible roles of CeD-/IBD-associated genetics in barrier dysfunction and epithelial immune function.

## 2. Intestinal Barrier Disfunction in CeD and IBD

IBD and CeD are both complex genetic disorders. IBD can be monogenic, as evidenced by a mutation in the *TTC7A* gene involved in maintaining epithelial apicobasal polarity and very early-onset IBD [7]. However, IBD and CeD are usually polygenic, meaning that disease-associated variants are located in coding or non-coding regions and can affect many different genes that contribute to disease. Examples of genes affected by mutations in protein-coding regions are *ATG16L1* [8], *NOD2* [9], and *INAVA/C1ORF106* [10]. *ATG16L1* and *NOD2* are important for barrier defense mechanisms, including the secretion of antimicrobial peptides in Paneth cells [11,12]. In Crohn’s disease (CD) patients, a variant in *ATG16L1* (rs2241880) is associated with impaired pathogen clearance, imbalanced cytokine production, and increased endoplasmic reticulum stress [13]. A mouse model study delineated the importance of this gene for intestinal epithelial cells (IECs): deleting *Atg16L1* in Paneth cells led to development of CD-like ileitis [11]. Moreover, loss-of-function of *NOD2* due to mutations in the coding region activates fibroblasts and macrophages, leading to fibrosis and a stricturing phenotype in CD [14]. IECs are connected by different barrier proteins like adherens junctions (AJs) and tight junctions (TJs) (Appendix B). Protein-coding SNPs in the CeD- and IBD-linked gene *C1ORF106* destabilize AJs, impairing barrier function [15,16].

Various loci—specific chromosomal regions of interest that may contain genes or regulatory elements associated with the disease—harbor non-coding SNPs that encompass genes implicated in barrier involvement in CeD and IBD, e.g., *PARD3*, *MAGI2* [17], and *CDH1* [18]. Intronic genetic variants in the TJ-regulating and cell polarity gene *PARD3* have been associated with CeD but not IBD [19]. Biopsies from CeD patients revealed altered Par-3 localization in IECs associated with an increased expression of pore-forming claudins and lowered the membrane localization of the TJ-regulating protein zonula occludens-1 [20]. The expression of MAGI2, an evolutionarily conserved scaffold protein found in TJ plaques, is reduced by a CeD-associated SNP in an intronic region, enhancing barrier permeability and gluten exposure [21]. Lastly, SNPs in intronic regions of *CDH1* (encoding E-cadherin) lead to splicing defects and a truncated form of E-cadherin [22]. Consequently, E-cadherin accumulates in the cytoplasm rather than at the plasma membrane in CD biopsies, contributing to defective barrier function [22].

These examples support a genetic contribution to barrier disfunction in CeD and IBD and imply that barrier disfunction may precede disease onset. Genetic effects on barrier function in both diseases may also be larger than previously thought. To date, most genes associated with immune-mediated diseases have been primarily studied in immune cells from peripheral blood or isolated from affected tissue. The advent of single-cell technologies now makes it possible to investigate the expression and function of disease-associated SNPs and genes in specific cell types, including epithelial cell populations. This will provide deeper insights into how particular genes contribute to intestinal barrier homeostasis and disfunction.

## 3. Genes in Shared CeD and IBD Loci Contribute to Intestinal Epithelial Barrier Function

GWASs in CeD and IBD have identified hundreds of disease-associated loci. For CeD, 43 loci besides HLA contribute to disease susceptibility. For IBD, 244 loci have been identified, and approximately 70% are shared between CD and ulcerative colitis (UC). In both diseases, most SNPs map near genes with a known immune function, e.g., in adaptive immune response and T- and B-cell stimulation [9,23,24].

To see which genes in CeD and IBD GWAS-associated loci may be implicated in epithelial cell function, we focused on genes in a 250Kb window around disease-associated SNPs shared between CeD and IBD (Appendix D). Notably, 206 genes in disease-associated loci are shared among CeD, CD, and UC (Figure 1A). According to the Gut Cell Atlas [25], 118 are expressed in IECs of the healthy human gut, suggesting that disease-associated SNPs might affect the regulation of these genes in an epithelium-specific fashion, contributing to disease pathology.

To identify pathways enriched in epithelial cells in CeD, CD, and UC, we applied over-representation analysis using the 118 expressed genes. Out of 118 genes, 38 contributed to the enrichment of specific pathways. The identified pathways and genes grouped into four immune-related clusters and one cluster involving the regulation of epithelial cell differentiation (Figure 1B). We summarized the differential expression status and identified *cis*-eQTLs of the 38 genes in the enriched pathways (Table 1), then reviewed the recent literature about the involvement of the most relevant genes in IBD and CeD.

### 3.1. IECs as Non-Conventional Antigen-Presenting Cells

The first cluster we identified (Figure 1B, in orange) is related to antigen processing and presentation and primarily includes HLA genes. In IBD and especially in CeD, HLA class II genes are strongly associated with disease. HLA genes are typically expressed in antigen-presenting cells, such as dendritic cells or B cells, to initiate an adaptive immune response. However, HLA genes are also expressed in IECs, which may function as non-conventional antigen-presenting cells using MHC class II molecules, and they are involved in CD4+ T effector activation, T regulatory activation, and epithelial stem cell renewal [36]. The overexpression of MHC class II in the IECs of CeD patients and HLA-DR upregulation in intestinal villi caused by gliadin were demonstrated decades ago [37]. The expression of MHC class II receptors on epithelial cells supports the notion that gliadin is presented to immune cells by IECs. Recently, Rahmani et al. [10] showed that MHCII-expressing IECs can activate gluten-specific CD4+ T cells upon the addition of gluten peptides, but they did not demonstrate a direct interaction between IECs and gluten-specific CD4+ T cells. Conversely, IEC-mediated gluten presentation to cytotoxic CD8+ intraepithelial lymphocytes (IELs) is unlikely as they have a T-cell receptor (TCR) repertoire that is distinct from gluten-specific CD4+ T cells [38]. These studies represent a paradigm shift in our thinking about the role of IECs in CeD, proving that they not only present specific antigens like gluten antigens alongside traditional antigen-presenting cells, thereby supporting activation of CD4+ T cells; they may also present other antigens to both CD4+ and CD8+ T cells, which might have additional roles in disease mechanisms.

Similarly, studies in IBD have shown that epithelial cells express HLA-DR and can activate CD4+ T cells in vitro, stimulating their proliferation and IFN-γ secretion [39,40,41,42]. Moreover, in vitro, colonic IECs can stimulate antigen-specific CD4+ T cells via MHC class II interactions and steer both regulatory and effector T-cell phenotypes following bacterial infection [43].

These studies highlight the importance of studying IECs as non-conventional antigen-presenting cells capable of activating and modulating the T-cell response. However, the consequences of non-conventional antigen-presentation by IECs in CeD and IBD have only just begun to be understood. Understanding how IECs promote tissue inflammation during disease requires further studies to elucidate which immune cells can be activated by IECs and which epitopes play a role.

### 3.2. Regulation of Immune Response in IECs

IECs can control immune responses in the intestine by producing cytokines and transducing signals from microbiota, influencing epithelial barrier homeostasis, the actixvation of IELs, and the production of cytokines and immunoglobulin [44,45].

The genes grouped in the “Regulation of immune response” pathway (Figure 1B, in blue) are mostly involved in T-cell recruitment and cytokine signaling. Genes of the tumor necrosis factor receptor superfamily (*TNFRSF14*, *TNFRSF1A*, and *LTBR*) are involved in apoptosis, inflammation, proliferation, survival, and differentiation [46]. *TNFRSF14* was shown to be involved in host defense against pathogenic bacteria, the prevention of inflammation, and protection against apoptosis during inflammation [47,48,49]. The absence of Tnfrsf14 in mice leads to accelerated and exacerbated colitis and impaired epithelial innate immunity [49]. Moreover, epithelial TNF receptors have been associated with promoting mucosal healing in UC, likely by enhancing Wnt/beta-catenin signaling via TNFRSF1A and TNFRSF1B [50].

Interleukin 10 (*IL10*) is an anti-inflammatory cytokine that suppresses antigen-presentation and the production of proinflammatory cytokines and chemokines. In the intestinal mucosa, IL-10 is produced by both immune and epithelial cells and is involved in maintenance of epithelial barrier integrity [51]. IL-10 is particularly relevant in IBD, as evidenced by the spontaneous development of colitis in *Il10*^-/-^ and *Il10Rb*^-/-^ mice [52] and the correlation of SNPs in *IL10* with the early onset of colitis [53]. Europeans who carry a T allele of SNP rs3024505, located in an enhancer near *IL-10*, have a higher risk of IBD [54]. If IECs (together with immune cells) represent a relevant source of IL-10 in the gut, variants that can downregulate *IL10* expression could lead to an imbalance between IL-10’s anti-inflammatory effects and the inflammatory effects of other cytokines such as IFN-γ and IL-21, contributing to the inflammation and barrier disruption observed in IBD.

Another gene in this group is gasdermin B (*GSDMB*). Gasdermins in IECs are important for intestinal homeostasis; they are involved in cell death and in forming pores in the cell membrane used for rapid cytokine release without leading to cell death [55,56]. The cleavage of GSDMB by granzyme A (GZMA) produced by natural killer (NK) cells and cytotoxic T cells triggers cell death via pyroptosis. *GSDMB* is also upregulated by IFN-γ, a key cytokine in CeD pathology [57], and in the IECs of IBD patients. Interestingly, many CeD- and IBD-associated SNPs have a *cis*-eQTL effect on *GSDMB* expression (Table 1), suggesting they could have a pivotal role in making IECs more prone to cell-killing via GZMA and cytokine release.

### 3.3. IECs Express T-Cell-Regulating Genes

The genes grouped under the “T-cell regulation” pathway (Figure 1B, in red) have well-documented functions in T cells, but it is unclear whether their expression in IECs contributes only to T-cell regulation or has additional roles in intestinal homeostasis. For instance, the loss-of-function of protein tyrosine phosphatase non-receptor type 2 (*PTPN2*) negatively regulates T-cell activation, increasing the risk of IBD [58] and CeD [59]. In CD patients, this loss-of-function results in high numbers of Th1 and Th17 cells and fewer Treg cells [60]. Interestingly, constitutive *Ptpn2* deficiency in mice causes the depletion and compromised functionality of Paneth and goblet cells [58,61], suggesting that this gene plays additional roles in IEC function.

Similarly, the expression of zinc finger MIZ-type containing 1 (*ZMIZ1*), which is involved in the TGF-beta signaling pathway and T-cell development [62,63], is increased in the inflamed gut tissue of IBD patients but not in CeD [64,65]. Likewise, in a colitis mouse model, *Zmiz1* mRNA and protein levels were dramatically increased in inflamed gut tissue, but not in blood [65]. This differential expression suggests a potential unique role for *ZMIZ1* in IECs in IBD pathology, although its exact function remains unclear.

Tyrosine kinase 2 (*TYK2*), a member of the JAK family involved in the signaling of cytokines, is a good example of a gene that has different roles depending on the cell type in which it is expressed. In a colitis mouse model, T cells lacking Tyk2 activity reduced Th1 cell differentiation and improved disease phenotype [66]. However, when IECs lack TYK2, they exacerbate disease by impeding IL-22/STAT3-dependent responses and impairing mucosal wound-healing processes [67]. TYK2 inhibitors are currently in clinical trials for IBD management [68], but their effect on IECs should be carefully evaluated.

*SH2B3*, also known as lymphocyte adaptor protein (LNK), is a negative regulator of the JAK-STAT, tyrosine kinase, and cytokine signaling pathways [24,69]. Along with its involvement in immune and inflammatory pathways, SH2B3 is involved in regulating integrin signaling, which is essential for cell migration and adhesion in endothelial cells and hematopoietic cells [70,71]. In the IECs of CeD patients, the *SH2B3* promoter is significantly hypomethylated, resulting in its overexpression compared to the healthy population [29]. Although SH2B3 is less studied in an epithelial cell context, *SH2B3* overexpression in IECs in CeD and its potential role in cellular microenvironment suggests SH2B3 might have a role in barrier function and CeD pathology.

These examples highlight that the genes known to regulate T cells may also play a crucial role in disease pathogenesis through barrier cells, as these genes are expressed in IECs. Changes in the expression of these genes in IECs might disturb their communication with different cell types, repair mechanisms, and other cellular processes, leading to disrupted gut homeostasis and subsequently to inflammation.

### 3.4. Interferon Signaling Genes in IECs

Interferon (IFN) responses are characteristic of both IBD and CeD, as highlighted by the “Interferon signalling” cluster in Figure 1B (in brown), and they may have pleiotropic effects. Excessive IFN-γ signaling exacerbates IBD through vascular barrier disruption, increasing the influx of immune cells [72]. Conversely, a recent investigation described a protective feature of IFN-γ signaling in IBD. In mice, when IECs were stimulated with IFN-γ and presented antigens to IELs, the latter produced ATPases that limited the accumulation of extracellular ATP. This prevented the activation of the NLRP3 inflammasome in tissue macrophages, limiting their production of proinflammatory cytokines and antigen presentation [73]. When the interferon-gamma receptor (*Ifngr*) was knocked out specifically in IECs, tissue macrophages produced IL-1α and IL-1β and presented antigens to CD4+ T cells, culminating in a proinflammatory response that leads to colitis [73]. Defective IFN-γ signaling in IECs may thus contribute to inflammation by reducing the protective feature of IFN-γ signaling.

*IRF1* (interferon regulatory factor 1) is a crucial transcription factor responsible for regulating IFN-γ-inducible genes. In general, IRF1 is involved in inflammation, immune response, cell proliferation and differentiation, programmed cell death, and the regulation of cell cycle [74]. In IECs, it has been implicated in the upregulation of IL-27, IL-15, and IL-7 expression [75,76], therefore playing a role in the activation and regulation of NK and T cells in the intestinal mucosa. IL-27 has also been shown to induce antigen processing and presentation via MHC class I and II IECs, via the STAT–IRF1–CIITA axis [77]. IL-27 is also involved in disruption of barrier integrity upon TNF-alpha stimulation, stimulating cell shedding [78]. IL-15, a proinflammatory cytokine, is overly expressed in the gut epithelium and lamina propria of both CeD and IBD and enhances T-cell activation and proliferation and proinflammatory cytokine production by both T cells and macrophages [79]. Lastly, IL-7 was recently shown to be critical for gluten-induced toxicity in CeD by upregulating NKG2C/D expression in CD8+ T cells [80].

*SOCS1* plays an important role in immune tolerance and the regulation of effector and regulatory T cells [81] and is a negative feedback inhibitor of JAK/STAT cytokine signal transduction [82]. A recent clinical case study revealed that *SOCS1* haploinsufficiency due to loss-of-function variants in *SOCS1* led to a broad range of intestinal complications and that JAK inhibitors can be a potential treatment strategy [83]. Another study of ten patients with various early-onset autoimmune diseases, including CeD, identified germline loss-of-function mutations in *SOCS1*. However, these studies focused on immune cells, and little is known about SOCS1’s function in epithelial cells, but it may promote apoptosis via p53 signaling after stimulation with IFN-γ [84].

The IFN-γ transcription factor STAT1 is expressed in nearly all epithelial cell types and can perform a dual role. During inflammation, or in response to epithelial damage, reserve intestinal stem cells (r-ISCs) located in the “+4” position proliferate and functionally contribute to intestinal regeneration, with the JAK/STAT1 pathway required for r-ISC activation [85]. Given the key role of IFN-γ in CeD, further investigation is needed to determine whether disease-associated SNPs in the *STAT1* locus contribute to crypt hyperplasia through the activation of the JAK/STAT1 pathway in r-ISCs. Conversely, along the villus axis, STAT1 can induce a complex cell death pathway in IECs as a defense mechanism against bacterial infection [86]. Overall, the role of the STAT1 pathway should be studied more deeply and in cell-specific contexts (e.g., mature enterocytes and stem cells) to tease apart its contributions to disease.

### 3.5. Genetic Impact on Intestinal Epithelial Cell Differentiation

Although most genes in shared CeD-IBD loci reviewed here are known to have immune functions, *NOTCH4*, *HES5*, and *SULT2B1* are known to be involved in the differentiation and maintenance of epithelial cells and hence in barrier function (as highlighted in the “Regulation of epithelial cell differentiation” cluster in Figure 1B, in pink).

The Notch signaling pathway is a well-known master regulator of cell differentiation and cell fate in the intestine [87]. NOTCH4 is a membrane receptor that, upon ligand activation, releases the Notch intracellular domain responsible for the transcriptional activation of genes involved in cell differentiation and proliferation [88]. *NOTCH4* SNPs rs9267835 and rs520692 have been associated with both CeD and IBD and confirmed using whole-exome sequencing. In CeD, rs9267835 is associated with increased disease risk, and rs520692 is associated with lower risk [89]. Since these SNPs are coding variants, they could impact NOTCH4’s ligand-binding and/or signal transduction. Moreover, the downregulation of Notch-signaling-related genes was observed in a mouse model of CD, along with decreased stem cell niche factors, increased intestinal crypt depth, and the impaired differentiation of absorptive lineages in the ileal epithelium [90]. These results suggest that *NOTCH4* genetic variants that limit the signaling function could result in defective Notch signaling and disruption of epithelial cell differentiation. Additionally, the transcription factor HES5, a downstream activator of Notch signaling, regulates IEC differentiation and proliferation in mice. Moreover, HES5 helps position stem cells correctly on the crypt–villous axis [91]. Notch signaling and HES5 may thus play a cooperative role in barrier homeostasis and recovery and in crypt hyperplasia in CeD.

Sulfotransferase family 2B member 1 (*SULT2B1*), responsible for the sulfation of cholesterol, is one of the main sources of cholesterol sulfate (CS) in the intestinal epithelium. CS is required for cholesterol biosynthesis, the proliferation of IECs, epidermal differentiation, platelet adhesion, and TCR signaling [92,93]. *SULT2B1* is downregulated in the small intestine of CeD patients and the ileum of CD patients (Table 1) [26,27] but upregulated in the colon of CD and UC patients, with CS reported to protect against DSS-induced colitis in mice [27,94]. These contrasting findings suggest that the direction of deregulation in *SULT2B1* expression depends on the disease and tissue affected. While inflamed cells in the colon seem to overexpress this gene to produce more CS and benefit from its protective effects, cells in the small intestine seem to lose this ability. Understanding the genetic regulation of *SULT2B1* in disease and how to restore normal expression may be an interesting approach to promoting IEC healing.

## 4. Conclusions and Future Perspectives

Linking genetics with the cell-type-specific expression of a gene potentially involved in disease is challenging. Thus far, most attention has focused on immune cells, as most genes associated with CeD and IBD are implicated in immune function, directly altering adaptive and innate immune responses. As emerging lines of evidence now implicate other cell types, including epithelial cells, we reviewed the known and putative functions of genes in disease-associated loci shared by CeD and IBD. By expressing genes involved in T-cell regulation, cytokine signaling, antigen presentation, and cell differentiation, IECs are active players in the disease mechanism. However, it is yet to be determined if the aberrant expression of these genes in IECs is the cause or consequence of inflammation (Figure 2).

To truly illustrate the direct genetic involvement of the epithelium in disease onset and exacerbation, cell-type-specific eQTL studies are needed. However, we currently lack large-scale eQTL or fine-mapping studies in IECs derived from patient materials. While fine-mapping causal SNPs and genes to a specific cell type remains difficult, recent advances in single-cell omics can help answer these questions. For example, Smillie et al. used single-cell RNA-seq to infer cell-type-specific risk genes for UC [40]. The authors showed that 29 of 57 GWAS-implicated genes are enriched in specific cell types and many epithelial cells, and there is a notably high number in M-like cells (Appendix B), suggesting that these cells play a greater role in UC than previously thought [40]. Similarly, Stankey et al. recently showed that the IBD risk allele of SNP rs2836882 is responsible for increasing *ETS2* expression in macrophages, resulting in exacerbated inflammation and upregulation of several IBD drug targets [68]. Applying similar approaches to IECs could elucidate their genetic contribution to CeD and IBD, offering insights into pathology and aiding in the search for better therapies through the fine-mapping of causal variants and genes.

Indeed, drugs developed considering genetic evidence are 2.6x more likely to be successful in clinical trials [95]. One study showed that 113 drugs with genetic evidence could potentially be used in IBD treatment because they target proteins that interact directly with a protein encoded by candidate risk genes [96]. Variants in *IL23R*, *JAK2*, and *TYK2* are strongly associated with IBD, highlighting the role of IL-23 signaling in disease pathogenesis. This understanding has contributed to the development of IL-23 inhibitors, such as ustekinumab and risankizumab, which have demonstrated efficacy in treating Crohn’s disease [97].

However, no currently approved treatments for IBD specifically target IECs, although they do promote mucosal healing [98]. To date, most clinical studies aimed at improving intestinal barrier function in IBD have focused on probiotic and prebiotic therapies, fecal microbiota transplantation (FMT), and mesenchymal stem cell (MSC) therapy and exome therapy [97]. In contrast, in CeD, IEC-directed therapies are gaining prominence, recognizing their central role in barrier function and immune modulation. For instance, larazotide acetate, a tight junction regulator, was tested to reduce intestinal permeability and immune activation, but it was discontinued at Phase 3 due to lack of efficacy [99,100]. A more recent approach, IMU-856, a SIRT6 activator, aims to promote epithelial repair in CeD without immunosuppression and has advanced to Phase 2 trials, showing potential for restoring barrier function [99]. Future therapies targeting IECs could leverage genetic insights to enhance efficacy. Treatments targeted at IECs to promote barrier integrity repair could specifically emphasize, for example, growth factors to stimulate IEC proliferation and differentiation, the increased production of TJ proteins, or the regulation of IECs’ production of pro- and anti-inflammatory cytokines.

To identify or repurpose drug targets to improve barrier function in CeD and IBD, functional studies are needed that link genetic targets to cell types to disease mechanisms. To do so, humanized models such as organ-on-a-chip [101] that account for patient genetics, disease heterogeneity, and disease-involved cell types offer a comprehensive framework to study these complex diseases. Looking ahead, the integration of these methodologies holds great potential for the development of targeted personalized treatments that considerably improve patient care and clinical outcomes.

## Figures and Tables

**Figure 1 ijms-26-02982-f001:**
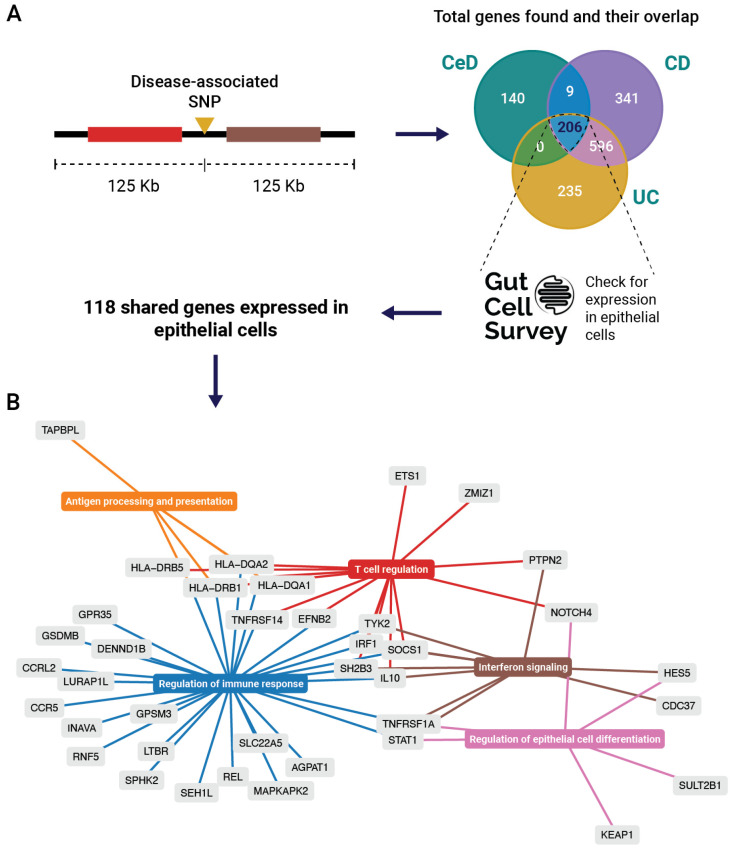
Overview of genes in shared loci of CeD, CD, and UC. (**A**) Selection of genes in disease-associated loci and numbers of overlapping genes across the three diseases. From the 206 overlapping genes, 118 genes are expressed in IECs according to the Gut Cell Atlas [25]. These 118 genes were used for over-representation analysis. (**B**) Over-representation analysis network plot showing the resulting groups of enriched pathways. Out of the 118 genes, only the genes present in the enriched pathways (*n* = 38) are depicted in the figure. More detailed information about the selection of these genes is in Appendix C.

**Figure 2 ijms-26-02982-f002:**
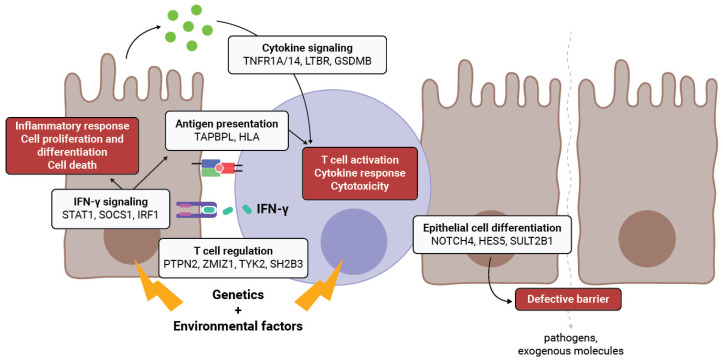
Roles of the genes shared between CeD and IBD in epithelial cells. Genes typically considered to have immune functions, e.g., antigen-presentation, T-cell regulation, and cytokine signaling, are also expressed in IECs. Epithelial cells therefore have the ability to modulate immune response and provide co-stimulation to immune cells. Under the right combination of genetics and environmental triggers, the expression of these genes or the function of these proteins can be altered in disease, promoting inflammation. Genes directly involved in IECs homeostasis and barrier integrity can also be affected by genetics and contribute to a defective barrier in disease.

**Table 1 ijms-26-02982-t001:** Differential expression and *cis*-eQTL status of genes in enriched pathways common to CeD and IBD.

Gene	Function ^1^	Differential Expression Status in Epithelial Cells		SNPs with *cis*-eQTL Effect	
		CeD ^2^	IBD ^3^	CeD	IBD
*TAPBPL*	Links MHC I molecules to TAP transporter				
*HLA-DQA2*	Antigen-presentation via MHC class II				
*HLA-DRB5*	Antigen-presentation via MHC class II	Down [26]	Up [25,27]	rs424232 [28]	rs3135005, rs693797, rs9267911 [28]
*HLA-DRB1*	Antigen-presentation via MHC class II		Up [27]		
*HLA-DQA1*	Antigen-presentation via MHC class II	Down [26]	Up (7)		rs3135005, rs9268403, rs693797 [28]
*TNFRSF14*	TNF receptor superfamily		Down [27]		
*IL10*	Immunoregulatory cytokine	Up [26]			
*ZMIZ1*	Transcriptional coactivator		Up [27]	rs1250567 [26]	
*ETS1*	Transcription factor				
*SH2B3*	Negative regulator of cytokine signaling	Down [26], Up [29]			
*EFNB2*	Ephrin receptor		Up [27]		
*SOCS1*	Negative regulator of interferon signaling		Up [27]		
*PTPN2*	Protein tyrosine phosphatase (signaling)		Down [27]		
*TYK2*	Tyrosine kinase (signaling)				
*IRF1*	Transcription factor of interferon genes	Up [26,30,31]	Up [27]		
*NOTCH4*	Membrane receptor of Notch signaling				
*CDC37*	Molecular chaperone		Down [27,32]		
*STAT1*	Signal transducer in response to interferon	Up [26,30,31]	Up [25,27,32]		
*HES5*	Transcription repressor				
*TNFRSF1A*	TNF receptor superfamily		Up [32]	rs2364484 [26]	
*DENND1B*	Regulates T-cell receptor internalization		Up [27]		
*INAVA/C1ORF106*	Cytokine production, adherens junctions				
*MAPKAPK2*	Stress-activated serine/threonine-protein kinase				
*SPHK2*	Catalyzes the phosphorylation of sphingosine				
*AGPAT1*	Converts lysophosphatidic acid into phosphatidic acid		Up [27]		
*GPSM3*	Regulator of GTPase activity		Up [32]		
*REL*	Proto-oncogene involved in lymphopoiesis		Down [27]		
*GSDMB*	Pore-forming protein		Up [27]	rs2305480, rs2872507, rs10852936, rs2305479, rs11557467, rs12950743, rs8067378 [28]	rs8069176, rs2305480, rs2872507, rs10852936, rs2305479, rs10445308, rs11557467, rs12950743, rs8067378, rs4795405, rs3902025, rs11078927 [28,33,34]
*SEH1L*	Component of nuclear pore complex				
*CCR5*	Beta chemokine receptor family				
*SLC22A5*	Cation transporter	Down [26]	Down [27]		
*RNF5*	Membrane-bound ubiquitin ligase		Down [27]		
*KEAP1*	Sensor of oxidative stress				
*SULT2B1*	Sulfation of cholesterol	Down [26]	Up (colon), Down (ileum) [27]		
*LTBR*	TNF receptor superfamily		Down [27]	rs10849448 [35], rs2364480, rs9669611, rs12354 [28], rs2364484 [26,28]	
*LURAP1L*	Regulation of NFkB signaling				
*GPR35*	G-protein coupled receptor	Down [26]	Down [27]		
*CCRL2*	Chemokine receptor				

^1^ According to GeneCards. ^2^ Bulk data from whole intestinal biopsies or intestinal epithelial layer. No single-cell or epithelial-only data was available for CeD when this manuscript was prepared. ^3^ Data from single-cell studies including epithelial cells.

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
