# Peer review of "Shared Genetics in Celiac Disease and Inflammatory Bowel Disease Specify a Greater Role for Intestinal Epithelial Cells"

_ijms, 2025, doi:10.3390/ijms26072982_

Round 1
Reviewer 1 Report
Comments and Suggestions for Authors
This study addresses the influence of genetics on the development of autoimmune intestinal diseases, such as celiac disease (CD) and inflammatory bowel disease (IBD). The article offers valuable insights into genetic mapping and its potential for developing more effective and targeted medications. However, I believe the manuscript could benefit from a more in-depth discussion, particularly regarding clinical studies that explore these diseases and the relationship between genetic factors and current treatments.
That said, the objectives of your work are clearly articulated, and I am confident that with some revisions, the manuscript will be suitable for acceptance.
Line 22: I suggest moving the citations [1,2] to the end of the sentence.
Line 29: I recommend removing the bold formatting from "Genome-Wide Association Studies (GWAS)" and other bolded words in this paragraph.
Lines 43-44: I suggest removing the bold formatting from the highlighted words and other bolded terms throughout the manuscript.
Line 66: The authors mention "loci." Could they clarify this term? It would be helpful to specify that it refers to the abbreviated notation indicating the chromosome number and the position of a specific gene. Please elaborate on this sentence.
Lines 78-86: Which references were used to support this paragraph? It would be beneficial to include them for clarity and credibility.
Table 1: I recommend that the authors revise Table 1 to align with the journal's formatting guidelines.
Author Response
Comment 1:
This study addresses the influence of genetics on the development of autoimmune intestinal diseases, such as celiac disease (CD) and inflammatory bowel disease (IBD). The article offers valuable insights into genetic mapping and its potential for developing more effective and targeted medications. However, I believe the manuscript could benefit from a more in-depth discussion, particularly regarding clinical studies that explore these diseases and the relationship between genetic factors and current treatments.
That said, the objectives of your work are clearly articulated, and I am confident that with some revisions, the manuscript will be suitable for acceptance.
Response: We appreciate the reviewer’s positive feedback on the significance of our study and the clarity of our objectives. We acknowledge the importance of discussing clinical studies that explore the relationship between genetic factors and current treatments for CeD and IBD. In response, we have expanded the discussion to provide a more comprehensive overview of clinical studies exploring the connection between genetic factors and current treatment approaches. We have also included additional insights into emerging therapies to improve disease management. We believe these revisions enhance the clinical relevance of our study and strengthen the discussion.
The changes are made in post table:1 line 184-206: Indeed, drugs developed considering genetic evidence are 2.6x more likely to be successful in clinical trials [95]. One study showed that 113 drugs with genetic evidence could potentially be used in IBD treatment because they target proteins that interact directly with a protein encoded by candidate risk genes [96]. Variants in IL23R, JAK2, and TYK2 are strongly associated with IBD, highlighting the role of IL-23 signaling in disease pathogenesis. This understanding has contributed to the development of IL-23 inhibitors, such as ustekinumab and risankizumab, which have demonstrated efficacy in treating Crohn’s disease [97].
However, no currently approved treatments for IBD specifically target IECs, although they do promote mucosal healing [98]. To date, most clinical studies aimed at improving intestinal barrier function in IBD focus on probiotic and prebiotic therapies, fecal microbiota transplantation (FMT), and mesenchymal stem cell (MSC) therapy and exome therapy [97]. In contrast, in CeD, IEC-directed therapies are gaining prominence, recognizing their central role in barrier function and immune modulation. For instance, larazotide acetate, a tight junction regulator, was tested to reduce intestinal permeability and immune activation, but was discontinued at Phase 3 due to lack of efficacy [99, 100]. A more recent approach, IMU-856, a SIRT6 activator, aims to promote epithelial repair in CeD without immunosuppression and has advanced to Phase 2 trials, showing potential for restoring barrier function [99]. Future therapies targeting IECs could leverage genetic insights to enhance efficacy. Treatments targeted at IECs to promote barrier integrity repair could specify emphasize on, for example, growth factors to stimulate IEC proliferation and differentiation, increased production of TJ proteins, or regulation of IEC production of pro- and anti-inflammatory cytokines.
Comment 2:
Line 22: I suggest moving the citations [1,2] to the end of the sentence.
Response: Thank you for the suggestion and we agree with your comment. We have moved the citation to the end of the sentence (line 24).
Comment 3:
Line 29: I recommend removing the bold formatting from "Genome-Wide Association Studies (GWAS)" and other bolded words in this paragraph.
Response: Agree. We have removed all the bold formations in that paragraph. The words are – “Genome-Wide Association Studies (GWAS) (line 29), single nucleotide polymorphisms (SNPs) (line30), promoters (line 34), enhancer(line 34)”.
Comment 4:
Lines 43-44: I suggest removing the bold formatting from the highlighted words and other bolded terms throughout the manuscript.
Response: Agree. We have removed all the bold formations in the main text. The words are – “dysbiosis (line 43), epitopes (line 44), monogenic (line 49), apicobasal polarity (line 50-51), ileitis (line 60), structuring (line 62), pyroptosis(line 20, Post table 1), Jak-STAT (line 50, Post table 1), NLRP3 inflammasome (line 71-72, Post table 1), haploinsufficiency (line 93, Post table 1), M-like cells (line 176, Post table 1)”.
Comment 5:
Line 66: The authors mention "loci." Could they clarify this term? It would be helpful to specify that it refers to the abbreviated notation indicating the chromosome number and the position of a specific gene. Please elaborate on this sentence.
Response: Thank you for pointing out the slightly confusing terminology. We agree with your suggestion, explaining the term will help the readers to understand how we have defined loci. We have made an additional explanation which is highlighted here: “Various loci—specific chromosomal regions of interest that may contain genes or regulatory elements associated with the disease—harbor non-coding SNPs that encompass genes implicated in barrier involvement in CeD and IBD, e.g. PARD3, MAGI2 [17], and CDH1 [18].” (line 66-69).
The term is further explained in the Appendix A (Glossary) (Line 257-260, post table 1): “Locus: (Plural: Loci), A genomic locus is a specific region of the genome that is defined based on an area of interest, such as a transcribed RNA, a single exon, or a region associated with differences in expression. It is typically chosen to encompass the feature of interest while minimizing the inclusion of unrelated genomic regions”.
Comment 6:
Lines 78-86: Which references were used to support this paragraph? It would be beneficial to include them for clarity and credibility.
Response: Thank you for your feedback. The paragraph in Lines 78–86 serves as a summary of the entire section, providing an overview and the authors’ perspective on the topic. As it does not introduce new data or specific claims, but rather synthesizes information discussed in the preceding text, no additional references were included. However, to enhance clarity, we have ensured that all key points summarized in this paragraph are properly referenced in the main discussion.
Comment 7:
Table 1: I recommend that the authors revise Table 1 to align with the journal's formatting guidelines.
Response: Agree. We have changed the format of Table 1 aligning the journal’s guideline. We have removed the background color of the table.
Reviewer 2 Report
Comments and Suggestions for Authors
This is a very well-written and balanced review, and is an easy read. Factually accurate with appropriate balance of conclusions and hypotheses. Organization is logical, conclusions are thought provoking. I do not detect bias. I do not have any suggestions to improve this article. Figures and tables are appropriate.
This is a review article, that explores the genes (and their putative functions) expressed in epithelial cells that are shared between CeD and IBD. Because most research in the field has focused on immune cells, intestinal epithelial cell research has been identified as a gap. This review provides insights starting from a genetic basis, into how barrier function and intestinal epithelial cells themselves could be important in both diseases, and hopefully stimulates more research into this area. They also note that to date, no drug therapy for IBD specifically targets intestinal epithelial cells. The authors found shared genes that enriched pathways related to immune activity and epithelial cell differentiation. Each pathway is reviewed in detail for the 38 relevant genes (shown in table 1). Authors highlight the need for large-scale eQTL or fine-mapping studies in IECs derived patient intestinal epithelium. They also point out that functional studies using technology such as the humanized “organ on chip” approach could be used to help identify drug targets for the intestinal epithelial cell. It's a call to action. 1) This topic is original to the field. There are no other review articles for this specific topic. It addresses a gap as noted above. 2) I do not have any specific suggestions regarding methodology specifically in Section 3 (portion of original contribution by authors), nor is this particular area within my expertise. 3) The references have been reviewed and are appropriate, even though some are quite old – reflects the limited advances in this particular area. 4) Figure 1 and 2, and Table 1 are appropriate and look good. 5) The statements, rationale are clear and present a convincing case that attempts to stimulate research in a new direction. This is well-written.
Author Response
Comments and Suggestions for Authors from Reviewer 2:
This is a very well-written and balanced review, and is an easy read. Factually accurate with appropriate balance of conclusions and hypotheses. Organization is logical, conclusions are thought provoking. I do not detect bias. I do not have any suggestions to improve this article. Figures and tables are appropriate.
This is a review article, that explores the genes (and their putative functions) expressed in epithelial cells that are shared between CeD and IBD. Because most research in the field has focused on immune cells, intestinal epithelial cell research has been identified as a gap. This review provides insights starting from a genetic basis, into how barrier function and intestinal epithelial cells themselves could be important in both diseases, and hopefully stimulates more research into this area. They also note that to date, no drug therapy for IBD specifically targets intestinal epithelial cells. The authors found shared genes that enriched pathways related to immune activity and epithelial cell differentiation. Each pathway is reviewed in detail for the 38 relevant genes (shown in table 1). Authors highlight the need for large-scale eQTL or fine-mapping studies in IECs derived patient intestinal epithelium. They also point out that functional studies using technology such as the humanized “organ on chip” approach could be used to help identify drug targets for the intestinal epithelial cell. It's a call to action. 1) This topic is original to the field. There are no other review articles for this specific topic. It addresses a gap as noted above. 2) I do not have any specific suggestions regarding methodology specifically in Section 3 (portion of original contribution by authors), nor is this particular area within my expertise. 3) The references have been reviewed and are appropriate, even though some are quite old – reflects the limited advances in this particular area. 4) Figure 1 and 2, and Table 1 are appropriate and look good. 5) The statements, rationale are clear and present a convincing case that attempts to stimulate research in a new direction. This is well-written.
Response: Thank you for your thoughtful and encouraging feedback. We are pleased that you found the manuscript well-written, balanced, and logically organized. We appreciate your recognition of the originality of the topic and the potential impact of our review in addressing the gap in the field.
We are glad that the references, figures, and tables met your expectations. Your positive comments on the clarity and rationale of the manuscript are much appreciated.
Round 2
Reviewer 1 Report
Comments and Suggestions for Authors
Dear authors,
Thank you for making the proposed changes. I believe that after this process the article has potential for publication. This review aims to map the genes and causal genes that may serve as targets for the development of new therapies to treat autoimmune diseases related to the intestine, such as celiac disease (CD) and inflammatory bowel diseases (IBD). In my opinion, this work has the potential to provide science and other researchers in the field with a valuable foundation for the development of new drugs, based on the review proposed here.